# Log Hyperbolic Cosine Loss Improves Variational Auto-Encoder

## Abstract

In Variational Auto-Encoder (VAE), the default choice of reconstruction loss function between the decoded sample and the input is the squared $L_2$. We propose to replace it with the log hyperbolic cosine (log-cosh) loss, which behaves as $L_2$ at small values and as $L_1$ at large values, and differentiable everywhere. Compared with $L_2$, the log-cosh loss improves the reconstruction without damaging the latent space optimization, thus automatically keeping a balance between the reconstruction and the generation. Extensive experiments on MNIST and CelebA datasets show that the log-cosh reconstruction loss significantly improves the performance of VAE and its variants in output quality, measured by sharpness and FID score. In addition, the gradient of the log-cosh is a simple tanh function, which makes the implementation of gradient descent as simple as adding one sentence in coding.

## 1 Introduction

Unsupervised generative modeling aims to represent probability distributions over input data. The explicit approach usually employs certain parametric model to estimate and approximate the unknown data distribution $p_{data}$, but the effect is sensitive to the model selection, for which expert knowledge is often needed. In a different vein, implicit generative models aim to generate samples from the data distribution without estimating the distribution itself. This greatly weaken the dependency on the model selection, and finds numerous applications such as simulating possible futures in reinforcement learning Finn & Levine (2017), predicting the next frame of a video sequence Lotter et al. (2015) and image super-resolution Ledig et al. (2017). Prominent examples of implicit generative models include Boltzmann machines Aarts & Korst (1988), Generative Adversarial Network (GAN) Goodfellow et al. (2014) and Variational Auto-Encoder (VAE) Kingma & Welling (2013). Among these methods, VAE enjoys an efficient and stable training process, and the encoder component of VAE gives dimension reduction and feature learning as a by-product. Meanwhile, it is also observed that images generated by VAE are often blurry. One potential reason is that the model fails to represent high-dimensional probability distributions accurately, which is the target issue we aim to address in this paper.

### 1.1 Variation Auto-Encoders

We now give a very brief account of VAE. In VAE, there is an encoder that converts the real input data $x$ to a latent code $z \sim q_\phi(z|x)$, and a decoder that reconstructs an image $\hat{x} \sim p_\theta(x|z)$ from the latent code $z$, where $q$ and $p$ are encoder and decoder, with parameter $\phi$ and $\theta$ respectively. VAE has two objectives: one is to match the decoded samples $\hat{x}$ to the input $x$, and the other is to maintain the posterior $q_\phi(z|x)$ of the hidden code vector $z$ to a given prior distribution $p(z)$. Combining these two, VAE uses the following loss function for datapoint $x$.

$$\mathcal{L}(\theta, \phi; x) = -\mathbb{E}_{q_\phi(z|x)} \log p_\theta(x|z) + D_{\mathrm{KL}}(q_\phi(z|x) \| p(z)). \tag{1}$$

Here the first term is the reconstruction loss, which reduces to the squared $L_2$ loss $\|x - \hat{x}\|_2^2$ if we assume that the decoder predicts a Gaussian distribution at each pixel i.e. $p_\theta(x|z) = \mathcal{N}(\hat{x}, \sigma^2 I)$. The second term of the objective matches the distribution of latent code to a prior $p(z)$, which by default is a standard normal distribution.

## 1.2 Reconstruction loss

To improve accuracy of representing probability distributions, many variants of VAE have been proposed. Much attention has been paid to improving the design of the second term, for example, changing the prior on the latent code (e.g. $\mathcal{S}$-VAE Davidson et al. (2018), VAE with householder flow Tomczak & Welling (2016)), and changing the loss between the latent distribution and the prior (e.g. AAE Makhzani et al. (2016), WAE Tolstikhin et al. (2018)).

An alternative approach for improvement is through the reconstruction loss, i.e. the first term in Eq. (1), which measures the error between the decoded sample and the input data points. Although the squared $L_2$ loss is widely used as reconstruction loss in VAE setting, it has long been observed that this loss function results in problems such as blurry images Mathieu et al. (2015). Several new metrics are proposed. Ridgeway et al. (2015) applied the structural-similarity (SSIM) index Wang et al. (2004) as a reconstruction metric of an auto-encoder. Larsen et al. (2016) employs a GAN discriminator to compute the reconstruction loss in VAE. Dosovitskiy & Brox (2016) combined a GAN discriminator with the perceptual loss which computes distances between image features extracted by deep neural networks. These experiments demonstrate the potential of improving VAE by re-designing the reconstruction loss. However, disadvantages are also evident: The first one Ridgeway et al. (2015) only demonstrated results on gray-scale images, which are very blurry. The last two Larsen et al. (2016); Dosovitskiy & Brox (2016) reported that their produce images suffer from severe high-frequency artifacts, possibly due to feature loss as discussed in Mahendran & Vedaldi (2015). In addition, as other GAN-based methods, introducing a discriminator makes the training harder and instable.

In this paper, we propose an element-wise reconstruction loss based on a simple log-cosh function

$$f(t; a) = \frac{1}{a} \log(\cosh(at)) = \frac{1}{a} \log \frac{e^{at} + e^{-at}}{2}, \tag{2}$$

where $a \in \mathcal{R}^+$ is a parameter and $\log$ is the natural logarithm. The plots of $f(t; a)$ in interval $[-1, 1]$ for different $a$ are shown in Fig. 1. The function is close to $L_1$ for large $|t|$ and close to $L_2$ for small $|t|$, thereby combining the smoothness advantage of $L_2$, and robustness and image sharpness advantage of $L_1$. In addition, the derivative of the function is simply the sigmoid function, rendering a very efficient training and very simple implementation.

In Sec. 4 we show empirically that even if the model trained with the log-cosh reconstruction loss achieved a larger error measured by $L_2$ loss, it significantly outperforms the model trained with squared $L_2$ loss. The result also indicates that squared $L_2$ loss can be ineffective in evaluating high-dimensional data like images, which will be discussed in more details later.

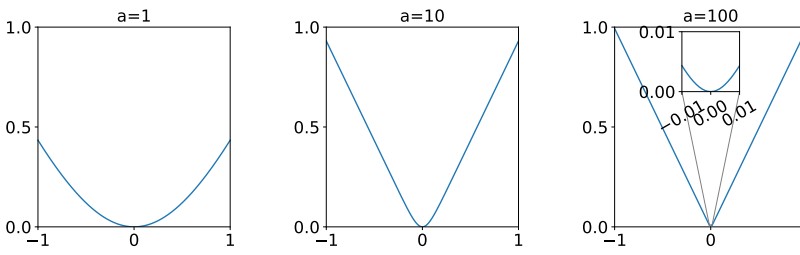

Figure 1: Plots of $f(t; a) = \frac{1}{a} \log(\cosh(at))$ with $a = 1, 10$ and $100$.

## 2 Limitations of traditional loss

Given two vectors $x$ and $\hat{x}$, the squared $L_2$ loss and the $L_1$ loss are defined as

$$\mathcal{L}_{\text{Squared L}_2}(x, \hat{x}) = \|x - \hat{x}\|_2^2 = \sum_i |x_i - \hat{x}_i|^2, \tag{3}$$

$$\mathcal{L}_{\text{L}_1}(x, \hat{x}) = \|x - \hat{x}\|_1 = \sum_i |x_i - \hat{x}_i|, \tag{4}$$

respectively, where $x_i$ stands for the $i$-th entry of $x$. The loss functions also extend to matrices by treating them as long vectors obtained from concatenating the rows. When applied in the VAE setting, $x$ and $\hat{x}$ are the input of encoder and output of decoder, respectively.

As mentioned in the last section, the squared $L_2$ loss widely used in VAE comes naturally from multivariate Gaussian distribution $p_\theta(x|z) \propto \exp(-\|x - \hat{x}\|_2^2)$. Other than the aforementioned issue of causing blurry images, the squared $L_2$ loss is also well known to be sensitive to large noise, and it has long been argued that in image space, it is ineffective in evaluating visual quality Wang & Bovik (2009) and makes the training to easily get stuck in local minima Zhao et al. (2017).

If we choose a decoder with zero-mean Laplace distribution $p_\theta(x|z) \propto \exp(-\|x - \hat{x}\|_1)$, we will obtain a loss function in terms of the $L_1$ norm. The $L_1$ loss is used as image to image loss in some variants of VAE and GAN, such as AEGAN Rosca et al. (2017) and cycle GAN Zhu et al. (2017). The gradient of $L_1$ loss is $\pm 1$ at differential points. The $L_1$ loss is generally robust to noise and, when used as a regularizer, encourages sparsity in solution. It is found to be quite useful in computer vision tasks such as deblurring Cai et al. (2009); Freeman et al. (2009); Xu & Jia (2010), since it appears to enable the solver to escape from local minima, although the exact reason for this is still elusive. In addition, Zhao et al. (2017) shows that $L_1$ can outperform $L_2$ when directly used as a loss function for neural networks in image restoration applications such as denoising Jain & Seung (2009), deblurring Xu et al. (2014), demosaicking Wang (2014), and super-resolution Dong et al. (2014). However, an obvious disadvantage of the $L_1$ loss is that it is not differentiable if any entry $x_i - \hat{x}_i = 0$, causing oscillation between $\pm 1$ during the training process.

In this paper, we will show that our proposed log-cosh loss takes the advantage of $L_1$ while overcoming the problem of non-differentiability. Experiments show that the log-cosh loss outperforms both $L_1$ and $L_2$ loss.

## 3 LOG-COSH LOSS

### 3.1 PROPERTIES OF LOG-COSH LOSS

Based on the log-cosh function shown in Eq. (2), we propose to use the log-cosh loss as the reconstruction error, taking advantages of both squared $L_2$ loss and $L_1$ loss while overcome their limitations. The log-cosh loss is defined as

$$\mathcal{L}_{\log\text{-}\cosh}(x, \hat{x}) = \sum_i f(x_i - \hat{x}_i, a) = \frac{1}{a} \sum_i \log(\cosh(a(x_i - \hat{x}_i))), \tag{5}$$

where $a \in \mathcal{R}^+$ is a hyper-parameter. We will first show that the log-cosh function behaves like $L_2$ around origin and like $L_1$ at other points.

**Proposition 1.** The log-cosh function $f(t; a)$ for $a > 0$ approximates $|t| - \frac{1}{a}\log 2$ when $t \to \infty$, and approximates $0.5at^2$ when $t \to 0$. More precisely,

$$f(t; a) = \frac{1}{a}\log(\cosh(at)) = \frac{1}{a}\log\frac{e^{at} + e^{-at}}{2} \to \begin{cases} |t| - \frac{1}{a}\log 2, & \text{when } |t| \to \infty \\ \\ 0.5at^2, & \text{when } |t| \to 0 \end{cases} \tag{6}$$

*Proof.* When $|t| \to \infty$, we have $f(t; a) = \frac{1}{a}\log((e^{a|t|} + e^{-a|t|})/2) \to \frac{1}{a}\log(e^{a|t|}/2) = |t| - \frac{1}{a}\log 2$. When $|t| \to 0$, the Taylor expansion gives $f(t; a) = 0.5at^2 + O(t^4)$. For general $t$, one has bounds $|t| - \frac{1}{a}\log 2 \leq f(t; a) \leq |t| - \frac{1}{a}\log 2 + \frac{1}{a}$. □

The log-cosh loss can be viewed as a smoothed version of $L_1$ loss differentiable everywhere. The gradient of $f(t; a)$ is

$$\frac{df(t; a)}{dt} = \tanh(at) = \frac{e^{at} - e^{-at}}{e^{at} + e^{-at}} = 2\sigma(2at) - 1, \tag{7}$$

where $\tanh(t) = (e^t - e^{-t})/(e^t + e^{-t})$ is the hyperbolic tangent function and $\sigma(t) = 1/(1 + e^{-t})$ is the sigmoid function, a commonly used activation function in deep networks.

### 3.2 MOTIVATION OF APPLYING LOG-COSH LOSS TO VAE

The log-cosh loss applied to VAE can be understood in the following way, which was also how we obtained the method. In general we hope to balance the reconstruction accuracy and latent space optimization in VAE. If the squared $L_2$ loss is used for the reconstruction, then the reconstruction accuracy deteriorates in the region of small reconstruction error, as $L_2$ penalizes these small errors too lightly, making the objection function Eq.(1) to be dominated by the second term (the KL divergence term). To solve this problem, simply increasing the relative weight of the reconstruction loss does not work well, since it harms the optimization in the latent space when the reconstruction error is large. Putting both large-error and small-error regions into consideration, a feasible solution is to increase the weight of reconstruction loss when it is small, but at the same time to clip the gradient of the loss with an upper bound so that the reconstruction penalty does not always increase linearly with the error. A smooth approximation of this is the tanh function, whose integral turns out to be the log-cosh function. The resulting log-cosh loss becomes more sensitive to small errors because it behaves as a $0.5a$-scaled squared $L_2$ loss around the origin. At the same time, it avoids penalizing too much when error is large when it behaves likes the $L_1$ loss. Therefore, we are motivated to implement the log-cosh loss to improve the reconstruction accuracy of VAE without damaging its latent space optimization.

The implementation of the log-cosh loss in VAE is quite simple, as the gradient is explicitly available as shown in Eq. (7). In tensorflow the implementation is as simple as replacing the reconstruction loss in the back-propagation with

$$\hat{\mathbf{x}} * \mathbf{tf.stop\_gradient}(\mathbf{2} * \mathbf{tf.sigmoid}(\mathbf{2a} * (\hat{\mathbf{x}} - \mathbf{x})) - \mathbf{1}),$$

where $x$ and $\hat{x}$ are the input of encoder and output of decoder, respectively. This form also avoids overflow or underflow during the computation, as it directly combines the gradient of the log-cosh loss (w.r.t decoder samples) and the gradient of the samples (w.r.t parameters).

## 4 EXPERIMENTS

We empirically compare the log-cosh loss to the squared $L_2$ loss in this section, leaving the comparison to $L_1$ to Appendix B. All the objective functions studied are explicitly given in Appendix A.1. The experiments are run on two datasets: (1) MNIST LeCun et al. (1998), which consists of 70k images of handwritten digits with size $28 \times 28$, and (2) CelebA Liu et al. (2015), which contains roughly 203k celebrity images cropped and resized to $64 \times 64$ as many previous works. We quantify the performance of VAE using two measures. The first one is sharpness, for which every image is convolved with the Laplace filter $\begin{pmatrix} 0 & 1 & 0 \\ 1 & -4 & 1 \\ 0 & 1 & 0 \end{pmatrix}$, and then the variance of the activations is computed as the sharpness score. The second one is Fréchet Inception Distance (FID) Heusel et al. (2017), for which the coding layer of a classification model trained by us is used to extract features and the Fréchet distance Fréchet (1957) is computed between distributions of the layer obtained from two groups of samples. FID captures the similarity of generated images to real ones in terms of vision-relevant features. We demonstrate that the log-cosh loss performs well on different architectures, including VAE Kingma & Welling (2013) and Wasserstein Auto-Encoders (WAE) Tolstikhin et al. (2018), with multi-layered perceptrons (MLP) networks and convolutional (Conv) networks.

### 4.1 EXPERIMENTAL SETUP

In all the experiments we use an Euclidian latent spaces $\mathcal{Z} = \mathcal{R}^d$ and Gaussian prior distributions $p_\theta(z) = \mathcal{N}(Z; \mathbf{0}, \mathbf{I}_d)$. Following Tolstikhin et al. (2018), we use $d = 8$ for MNIST and $d = 64$ for CelebA. For MNIST, we report results for MLP architecture and Conv architecture. For CelebA, we only use convolutional networks, considering that VAE implemented with MLP always performs badly on complex images like human faces. In all models, we use the sigmoid function at the output layer of the decoder.

We test the quality of reconstructed samples and generated samples. The reconstructed samples are the outputs of the decoder given vectors $z$ encoded from input image samples. The generated samples are obtained by decoding random noise vectors $z$ sampled from standard Gaussian. The sharpness we obtain is averaged across $10^3$ randomly selected images, and the FID is computed

Table 1: Sharpness (larger is better) and FID (smaller is better) on MNIST dataset.

| Algorithm | | Sharpness | | FID |
|---|---|---|---|---|
| | | Reconstructed Samples | Generated Samples | |
| VAE (MLP) | Squared $L_2$ | 0.052 | 0.040 | 200 |
| | Log-cosh | 0.168 **(+223%)** | 0.138 **(+245%)** | 65 **(-68%)** |
| VAE (Conv) | Squared $L_2$ | 0.068 | 0.063 | 114 |
| | Log-cosh | 0.156 **(+129%)** | 0.146 **(+132%)** | 25 **(-78%)** |
| WAE (MLP) | Squared $L_2$ | 0.082 | 0.078 | 101 |
| | Log-cosh | 0.158 **(+93%)** | 0.149 **(+91%)** | 35 **(-65%)** |
| WAE (Conv) | Squared $L_2$ | 0.111 | 0.100 | 56 |
| | Log-cosh | 0.166 **(+50%)** | 0.159 **(+59%)** | 24 **(-57%)** |

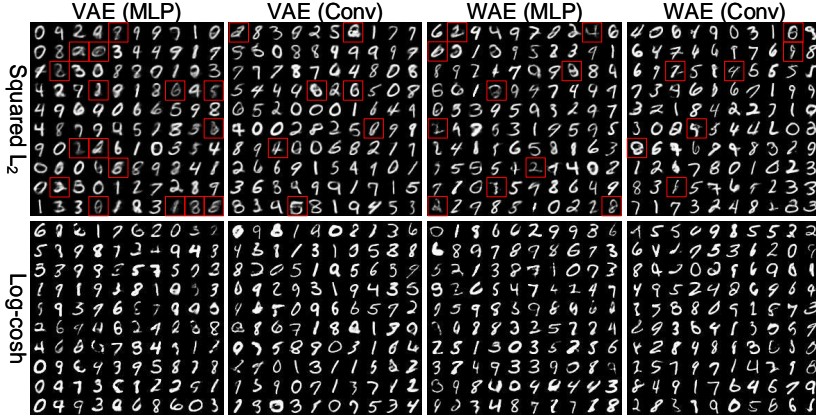

Figure 2: Random generated samples of VAEs trained on MNIST dataset.

based on $10^4$ randomly generated samples. For further details on experiment setup, please refer to Appendix A.

## 4.2 RESULTS

Numerical results on MNIST dataset are summarized in Table. 1, from which we can see that the log-cosh loss improves the sharpness up to $245\%$ and the FID up to $78\%$. In all models, the log-cosh loss consistently outperforms the squared $L_2$ loss. Larger sharpness stands for images of higher quality while smaller FID means that the distribution of generated samples is closer to the distribution of real dataset in terms of visual-relevant features. One can also observe that WAE improves sample quality compared with plain VAE, but the log-cosh loss can further improve WAE. We show generated samples in Fig. 2. As can be seen, many samples generated by traditional methods are quite blurry; some blurry examples are highlighted using colored boxes. After replacing the squared $L_2$ loss with our log-cosh function, we get sharper examples with a clearer background.

The reconstructed samples are shown in Fig. 3, which further confirms the benefit of the log-cosh loss. Our method is able to reconstruct sharp samples accurately. We use boxes to highlight some corresponding samples that are either wrongly reconstructed or extremely blurry when the squared $L_2$ loss function is used. We also conduct interpolation experiments, i.e., linearly interpolate between latent vectors and decode samples from these vectors. The interpolation results are reported in Appendix C, and suggest that our method enforces well-behaved manifolds in the latent space.

Results on CelebA dataset are summarized in Table. 2. In VAE, the log-cosh loss outperforms the squared $L_2$ loss by a factor of $\approx 30\%$ in both sharpness and FID. In WAE, the sharpness does not improve much, while the FID score is significantly improved by $30\%$. This indicates that our method better approximates the real dataset distribution. Generated samples shown in Fig. 4 further confirms this: Some samples of traditional methods are very dark and blurry, and some suffer from serious artifacts as marked by boxes. In comparison, our method generates many fewer such examples.

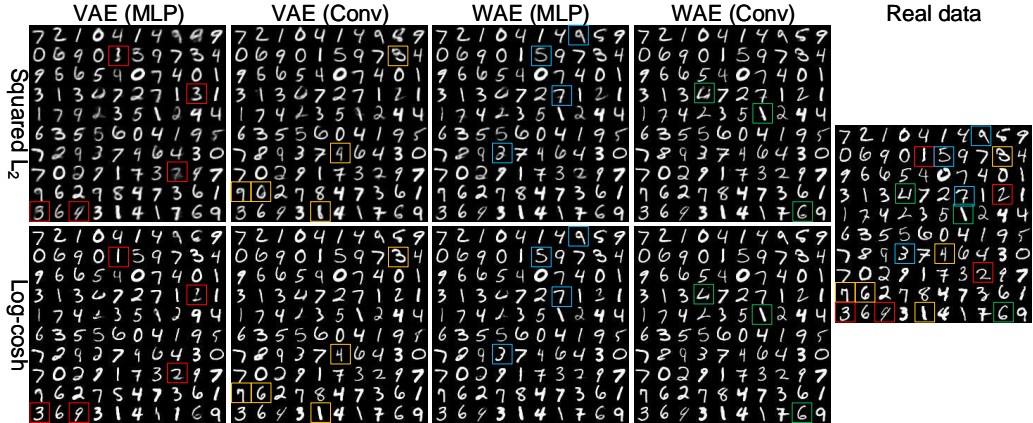

Figure 3: Reconstructed samples of VAEs trained on MNIST dataset. For comparision, some corresponding samples are marked by boxes with the same color.

Table 2: Sharpness (larger is better) and FID (smaller is better) on CelebA dataset.

| Algorithm | | Sharpness | | FID |
|---|---|---|---|---|
| | | Reconstructed Samples | Genarated Samples | |
| VAE (Conv) | Squared $L_2$ | $3.1 \times 10^{-3}$ | $2.3 \times 10^{-3}$ | 46 |
| | Log-cosh | $4.1 \times 10^{-3}$ **(+32%)** | $3.0 \times 10^{-3}$ **(30%)** | 31 **(-33%)** |
| WAE (Conv) | Squared $L_2$ | $6.7 \times 10^{-3}$ | $5.8 \times 10^{-3}$ | 30 |
| | Log-cosh | $7.0 \times 10^{-3}$ **(+4%)** | $5.7 \times 10^{-3}$ **(-2%)** | 21 **(-30%)** |

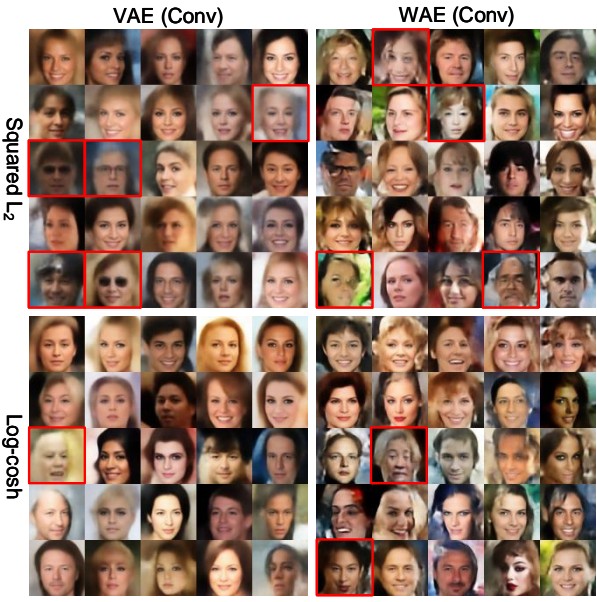

Figure 4: Random generated samples of VAEs trained on CelebA dataset. Samples that are very dark and blurry or suffer from serious artifacts are highlighted.

In Fig. 5 we show the reconstruction results, and we find our method to reconstruct input images more accurately. For example, the images reconstructed by traditional method and marked by boxes have either a wrong facial expression or a wrong gender, while our method reconstructs these features accurately.

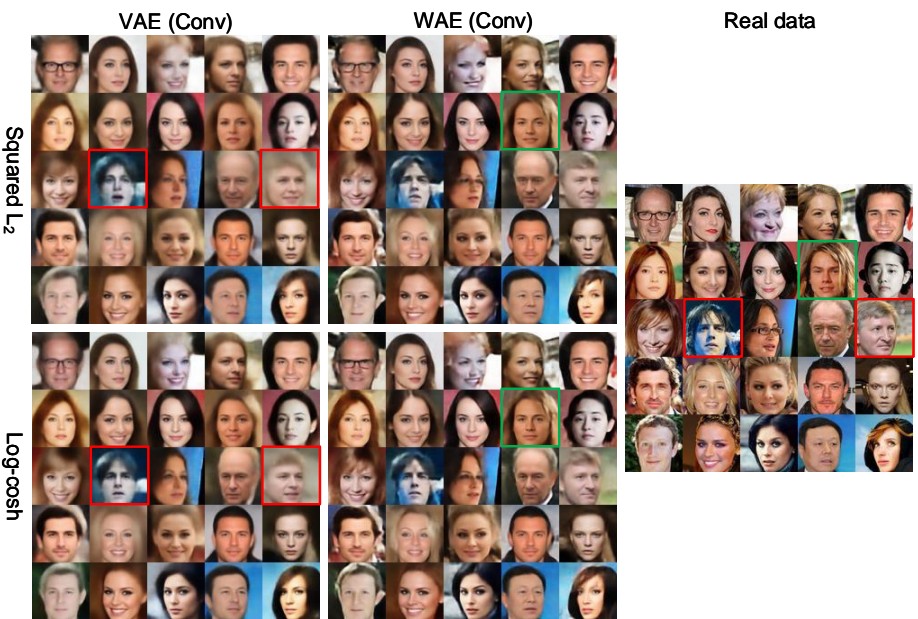

Figure 5: Reconstructed samples of VAEs trained on CelebA dataset. For comparision, some corresponding samples are marked by boxes with the same color.

### 4.3 DISCUSSION

In this section, we present a more in-depth analysis of the results shown in Sec. 4.2.

#### 4.3.1 IS $L_2$ A GOOD DISTANCE MEASURE FOR SIMILAR PICTURES?

We study the titled question in the context of VAE by examining the $L_2$ and log-cosh loss functions. Take VAE (MLP) as an example, and use the squared $L_2$ and the log-cosh as loss functions. We keep track of the quality of the parametrized networks $N_{L_2}^i$ and $N_{log\text{-}cosh}^i$ after each mini-batch $i$ during the training. Here the quality is measured by the squared $L_2$ loss when running the networks on the testing set, and this quality comparison is recorded in Fig. 6(a). We can see that both training processes converge, and the $L_2$ loss of the $N_{L_2}^i$ networks is consistently smaller than that of the $N_{log\text{-}cosh}^i$ networks. However, as we have seen from both the sharpness/FID scores and visual comparison in Sec. 4.2, the $N_{log\text{-}cosh}^i$ networks actually generate better figures than the $N_{L_2}^i$ networks. We also showcase one specific example in the figure, where two pictures of digit 3 are reconstructed. The upper one is clearly better, but it has a larger $L_2$ error. This suggests that the $L_2$ loss function is not a good measure for similar pictures in the context of VAE.

#### 4.3.2 BALANCE BETWEEN RECONSTRUCTION AND LATENT SPACE

In this section we explain that the log-cosh loss contributes to finding a better balance between the reconstruction and the latent space optimization, thus improving the performance of VAE. Recall that VAE simultaneously minimizes two losses: the reconstruction loss which measures the error between input of encoder and output of decoder, and the latent space loss which matches the distribution of latent vectors to a given prior distribution. The two losses are conflicting each other, and decreasing one often causes the other to increase. The trade-off between reconstruction and latent space is crucial because the goal is to accurately reconstruct the data points and at the same time to obtain a compact manifold on latent space for generation.

We show in Sec. 3 that the log-cosh loss behaves like a $0.5a$-scaled squared $L_2$ loss at small errors, and like a $L_1$ loss at large errors. When the reconstruction error is small, a large $a$ makes it visible in the objective function and in turn further decreases the reconstruction error. When the error is large, the gradient of log-cosh loss is bounded by $L_1$, which avoids harming latent space much. We

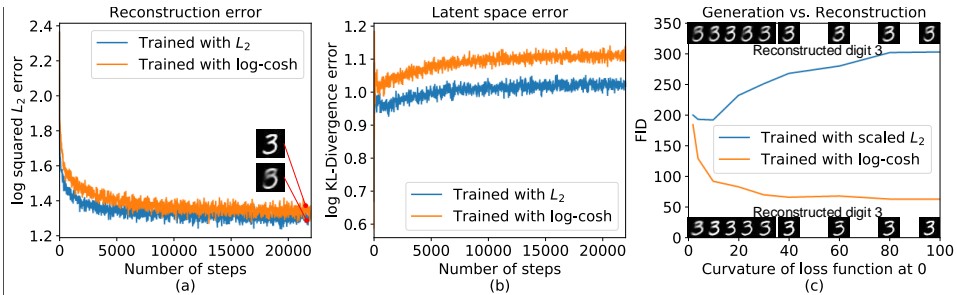

Figure 6: (a) Reconstruction error during training. For comparison, we use squared $L_2$ to measure the error of both models. We show two corresponding reconstructed sample here: While the upper one is clearly better, it has a larger $L_2$ error. (b) Latent space error measured by $D_{\mathrm{KL}}(q_\phi(z|x) \parallel p_\theta(z))$. (c) Generation quality measured by FID (smaller is better) of models trained with log-cosh and $\lambda$-scaled squared $L_2$ loss. We show reconstructed digit 3 of models trained with $L_2$ on the top and log-cosh on the bottom. The curvature at 0 is $a$ for log-cosh with parameter $a$ and $2\lambda$ for $\lambda$-scaled squared $L_2$. Since the log-cosh loss behaves like a $0.5a$-scaled squared $L_2$ loss only when the error is small and is not sensitive to large outliers, it does not harm optimization of latent space much and increasing $a$ improves reconstruction and generation simultaneously. In contrast, without our loss, simply scaling the $L_2$ does improve reconstruction but severely harm generation. Moreover, the reconstructed samples are still worse than our method.

show in Fig. 6(b) that the latent space error of model trained with the log-cosh loss ($a = 100$) does not increase much compared with the one trained with $L_2$. To further investigate the relationship between reconstruction and latent space, we train models with increasing parameter $a$ and draw the reconstructed digits and FID of generation in Fig. 6(c), which shows that our method improves reconstruction and generation simultaneously. This confirms that the log-cosh loss improves reconstruction without breaking its balance with latent space optimization. In contrast, without our log-cosh loss, the $\lambda$-scaled squared $L_2$ does improve the reconstruction sometimes with increasing $\lambda$. However, the scaled $L_2$ is quite sensitive to outliers, which harms optimization of latent space especially when the reconstruction error is large, thus severely decreasing generation quality as shown in Fig. 6(c).

### 4.3.3 MORE APPLICATIONS AND FUTURE WORK

The advantages of the log-cosh loss should make it useful in more applications. We provide an example of text removal in Appendix D, where the log-cosh loss consistently outperforms the $L_2$ and $L_1$ loss. It is worth trying to apply the log-cosh loss to various image restoration tasks where $L_1$ can outperform $L_2$, as demonstrated by Zhao et al. (2017). More tasks worth exploring include denoising Jain & Seung (2009), deblurring Xu et al. (2014), demosaicking Wang (2014), super-resolution Dong et al. (2014), etc.

## 5 CONCLUSION

In this paper, we propose to use the log-cosh loss as the reconstruction error of VAE.[1] We provide theoretical justifications, and conduct extensive experiments. Empirical results demonstrate that compared with traditional $L_2$ loss, the log-cosh loss improves the reconstruction without damaging the latent space optimization, thus automatically keeping a balance between the reconstruction and the generation. As a result, our method significantly improves the performance of VAE on image data in terms of sharpness and FID score. We also show the usefulness of the log-cosh loss for text removal, and call for more systematic studies of the loss function on other tasks in vision and general machine learning problems.

---

[1]We also propose to use the $L_1$ loss for special image cases in which most pixels of input images are very close to 0 or 1; see Appendix B for detailed discussions on this.

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

## A  FURTHER DETAILS ON EXPERIMENTS

### A.1  OBJECTIVE FUNCTIONS

In all implementations, we just change the reconstruction loss term while keep the loss term of latent space unchanged. We list all objective functions here, including VAE and WAE with the squared $L_2$, $L_1$ and log-cosh loss as the reconstruction loss.

$$\mathcal{L}_{\text{VAE-L}_2}(\theta, \phi; x) = \lambda \cdot \mathbb{E}_{\substack{z \sim q_\phi(z|x) \\ \hat{x} \sim p_\theta(x|z)}} \left[ \|x - \hat{x}\|_2^2 \right] + D_{\text{KL}}(q_\phi(z|x) \| p(z))$$

$$\mathcal{L}_{\text{VAE-L}_1}(\theta, \phi; x) = \lambda \cdot \mathbb{E}_{\substack{z \sim q_\phi(z|x) \\ \hat{x} \sim p_\theta(x|z)}} \left[ \|x - \hat{x}\|_1 \right] + D_{\text{KL}}(q_\phi(z|x) \| p(z))$$

$$\mathcal{L}_{\text{VAE-log-cosh}}(\theta, \phi; x) = \lambda \cdot \mathbb{E}_{\substack{z \sim q_\phi(z|x) \\ \hat{x} \sim p_\theta(x|z)}} \left[ \mathcal{L}_{\text{log-cosh}}(x, \hat{x}) \right] + D_{\text{KL}}(q_\phi(z|x) \| p(z))$$

$$\mathcal{L}_{\text{WAE-L}_2}(\theta, \phi; x) = \lambda \cdot \mathbb{E}_{\substack{z \sim q_\phi(z|x) \\ \hat{x} \sim p_\theta(x|z)}} \left[ \|x - \hat{x}\|_2^2 \right] + D_{\text{MMD}}(q_\phi(z|x) \| p(z))$$

$$\mathcal{L}_{\text{WAE-L}_1}(\theta, \phi; x) = \lambda \cdot \mathbb{E}_{\substack{z \sim q_\phi(z|x) \\ \hat{x} \sim p_\theta(x|z)}} \left[ \|x - \hat{x}\|_1 \right] + D_{\text{MMD}}(q_\phi(z|x) \| p(z))$$

$$\mathcal{L}_{\text{WAE-log-cosh}}(\theta, \phi; x) = \lambda \cdot \mathbb{E}_{\substack{z \sim q_\phi(z|x) \\ \hat{x} \sim p_\theta(x|z)}} \left[ \mathcal{L}_{\text{log-cosh}}(x, \hat{x}) \right] + D_{\text{MMD}}(q_\phi(z|x) \| p(z))$$

$$(8)$$

where $\lambda > 0$ is a hyperparameter which by default equals to 1 in the context of VAE, the $\mathcal{L}_{\text{log-cosh}}$ is the log-cosh loss defined in Eq. (5), $D_{\text{KL}}$ is the KL-divergence and $D_{\text{MMD}}$ is a divergence called the *maximum mean discrepancy* Tolstikhin et al. (2018). To be accurately, the loss function of WAE should be given w.r.t batches of $x$ and now the second term is $D_{\text{MMD}}(\int_x q_\phi(z|x)p(x)dx \| p(z))$.

### A.2  MNIST

In all experiments on MNIST, we use mini-batches of size 100 and train the model for 40 epochs. The activation function is Rectified Linear Unit (ReLU). All models are trained by Adam with the learning rate $\eta = 10^{-3}$. The log-cosh reconstruction loss Eq.(2) is implemented with parameter $a = 100$.

VAE (MLP) and WAE (MLP) use the relatively simple neural networks of multi-layered perceptrons (MLP). The encoder architecture is: $x \in \mathcal{R}^{28 \times 28} \rightarrow FC_{500} \rightarrow FC_{500} \rightarrow FC_8$, and the decoder architecture is: $z \in \mathcal{R}^8 \rightarrow FC_{500} \rightarrow FC_{500} \rightarrow FC_{28 \times 28}$.

In the VAE (Conv) and WAE (Conv), we use convolutional architectures with $4 \times 4$ convolutional filters and batch normalization Ioffe & Szegedy (2015). The encoder architecture is: $x \in \mathcal{R}^{28 \times 28} \rightarrow Conv_{128} \rightarrow Conv_{256} \rightarrow Conv_{512} \rightarrow Conv_{1024} \rightarrow FC_8$, and the decoder architecture is: $z \in \mathcal{R}^8 \rightarrow FC_{7 \times 7 \times 1024} \rightarrow Deconv_{512} \rightarrow Deconv_{256} \rightarrow Deconv_1$.

Here the $Conv_k$ stands for convolution with $k$ filters, the $Deconv_k$ stands for deconvolution with $k$ filters, and the $FC_k$ stands for fully connected layer with $k$ neurons.

### A.3  CELEBA

In all experiments on CelebA, we use mini-batch size 100, ReLU activation, and Adam training algorithm as the MINST experiment. For the learning rate, we follow WAE Tolstikhin et al. (2018) by initializing it to $\eta = 10^{-3}$ and decreasing it by factor of 2 after 30 epochs, and further by factor of 5 after first 50 epochs. The reported VAE (Conv) was trained for 68 epochs and WAE (Conv) for 55 epochs. We use $a = 100$ in VAE (Conv) and $a = 2.5$ in WAE (Conv).

We use convolutional architectures with $4 \times 4$ convolutional filters and batch normalization Ioffe & Szegedy (2015). The encoder architecture is: $x \in \mathcal{R}^{64 \times 64 \times 3} \rightarrow Conv_{128} \rightarrow Conv_{256} \rightarrow Conv_{512} \rightarrow Conv_{1024} \rightarrow FC_{64}$, and the decoder architecture is: $z \in \mathcal{R}^{64} \rightarrow FC_{8 \times 8 \times 1024} \rightarrow Deconv_{512} \rightarrow Deconv_{256} \rightarrow Deconv_{128} \rightarrow Deconv_3$.

## B  EXPERIMENTAL RESULTS OF $L_1$ LOSS

In this section, we test the $L_1$ loss on MNIST and CelebA dataset, and compare the numerical results with the $L_2$ loss and the log-cosh loss reported in Sec. 4. Samples of model trained with

Table 3: Sharpness (larger is better) and FID (smaller is better) on MNIST dataset. All the percentages are computed w.r.t squared $L_2$ loss.

| Algorithm | | Sharpness | | FID |
|---|---|---|---|---|
| | | Reconstructed Samples | Genarated Samples | |
| VAE (MLP) | Squared $L_2$ | 0.052 | 0.040 | 200 |
| | Log-cosh | 0.168 **(+223%)** | 0.138 **(+245%)** | 65 **(-68%)** |
| | $L_1$ | 0.175 **(+237%)** | 0.149 **(+273%)** | 61 **(-70%)** |
| VAE (Conv) | Squared $L_2$ | 0.068 | 0.063 | 114 |
| | Log-cosh | 0.156 **(+129%)** | 0.146 **(+132%)** | 25 **(-78%)** |
| | $L_1$ | 0.162 **(+138%)** | 0.154 **(+144%)** | 22 **(-81%)** |
| WAE (MLP) | Squared $L_2$ | 0.082 | 0.078 | 101 |
| | Log-cosh | 0.158 **(+93%)** | 0.149 **(+91%)** | 35 **(-65%)** |
| | $L_1$ | 0.162 **(+98%)** | 0.155 **(+99%)** | 37 **(-63%)** |
| WAE (Conv) | Squared $L_2$ | 0.111 | 0.100 | 56 |
| | Log-cosh | 0.166 **(+50%)** | 0.159 **(+59%)** | 24 **(-57%)** |
| | $L_1$ | 0.161 **(+45%)** | 0.152 **(+52%)** | 31 **(-45%)** |

Table 4: Sharpness (larger is better) and FID (smaller is better) on CelebA dataset. All the percentages are computed w.r.t squared $L_2$ loss.

| Algorithm | | Sharpness | | FID |
|---|---|---|---|---|
| | | Reconstructed Samples | Genarated Samples | |
| VAE (Conv) | Squared $L_2$ | $3.1 \times 10^{-3}$ | $2.3 \times 10^{-3}$ | 46 |
| | Log-cosh | $4.1 \times 10^{-3}$ **(+32%)** | $3.0 \times 10^{-3}$ **(30%)** | 31 **(-33%)** |
| | $L_1$ | $4.0 \times 10^{-3}$ **(+29%)** | $3.0 \times 10^{-3}$ **(+30%)** | 39 **(-15%)** |
| WAE (Conv) | Squared $L_2$ | $6.7 \times 10^{-3}$ | $5.8 \times 10^{-3}$ | 30 |
| | Log-cosh | $7.0 \times 10^{-3}$ **(+4%)** | $5.7 \times 10^{-3}$ **(-2%)** | 21 **(-30%)** |
| | $L_1$ | $6.5 \times 10^{-3}$ **(-3%)** | $5.1 \times 10^{-3}$ **(-12%)** | 33 **(+10%)** |

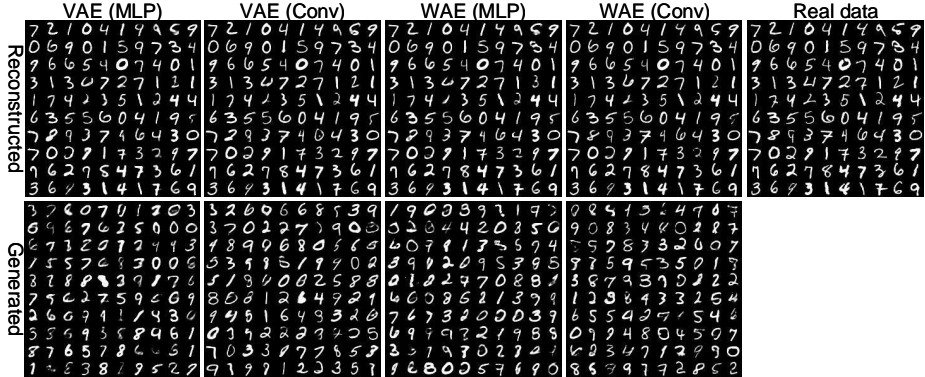

Figure 7: Samples of VAEs trained with $L_1$ reconstruction loss on MNIST dataset.

$L_1$ reconstruction loss are shown in Fig 7 and Fig. 8, and the numerical results are summarized in Table 3 and Table 4, where the improvement percentages are computed w.r.t the squared $L_2$ loss.

One can see that on the MNIST dataset, the log-cosh loss performs much better than the $L_2$ loss, as expected. It is also observed that the $L_1$ loss also performs very well, and sometimes even slightly better than the log-cosh loss. This is because 90% pixels of MNIST data are very close to binary value $\{0, 1\}$ and the outputs of our networks are in the interval $(0, 1)$, always falling into one side of the ground truth value (0 or 1). Thus the typical issue of the $L_1$ loss oscillating between non-differential point does not occur often here, and thus the advantage of log-cosh being smooth around 0 does not exhibit benefits than $L_1$.

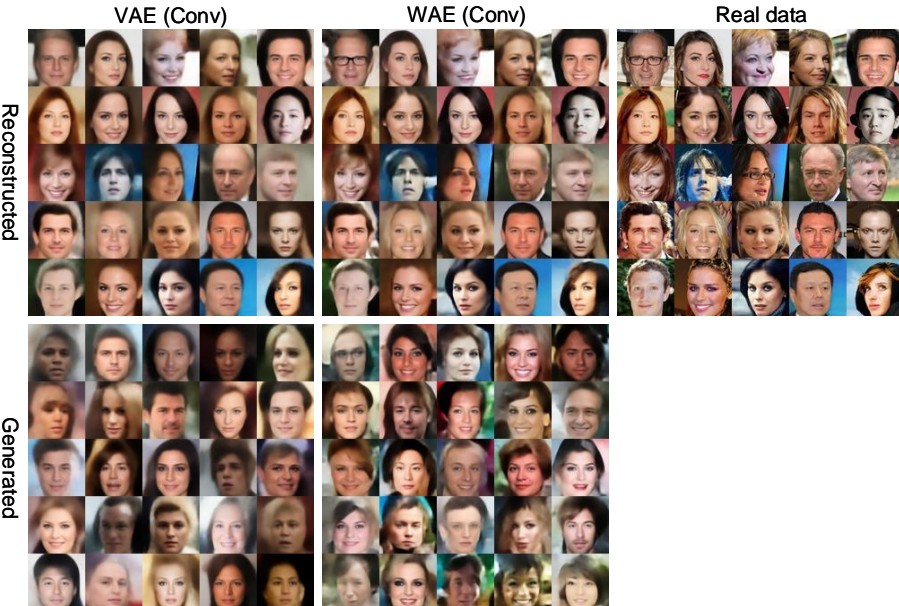

Figure 8: Samples of VAEs trained with $L_1$ reconstruction loss on CelebA dataset.

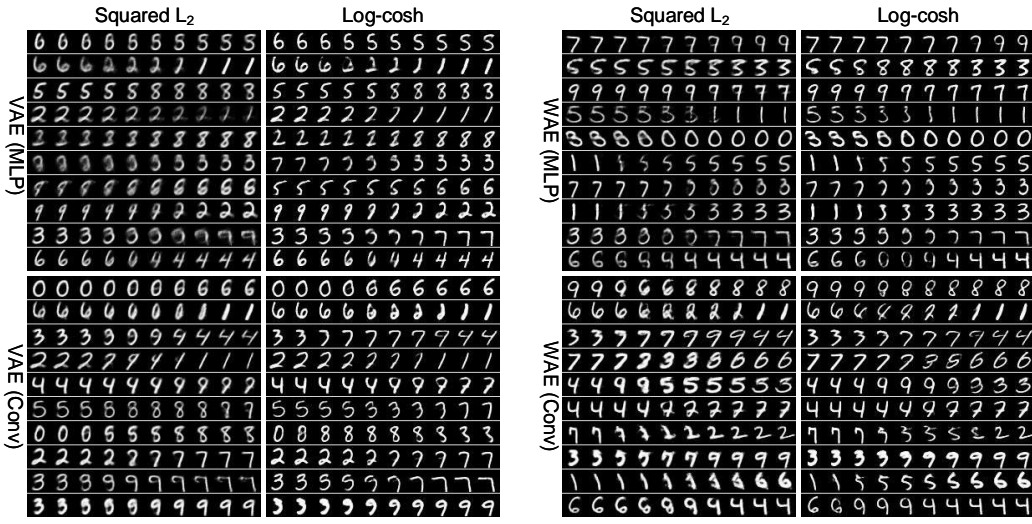

Figure 9: Interpolation results of VAEs trained on MNIST.

In comparison, for colored images or black-white images with gray pixels, the ground truth in general is somewhere between 0 and 1, and an output of a network may fall left or right of this value, suffering from the oscillation of derivatives of $L_1$. Indeed, we can see that on CelebA, the $L_1$ loss performs much worse than the log-cosh loss in both scores (Table 4) and the visualization (Fig. 8).

## C  INTERPOLATION EXPERIMENTS ON MNIST

We randomly take two samples $(x_1, x_2)$ from real dataset and feed them to the encoder to get the latent vectors $(z_1, z_2)$. Then we linearly interpolate between $z_1$ and $z_2$ in the latent space and use them to decode images. The decoded images are shown in Fig 9. The log-cosh loss enforces well behaved manifold in latent space, which is obviously better than squared $L_2$ loss.

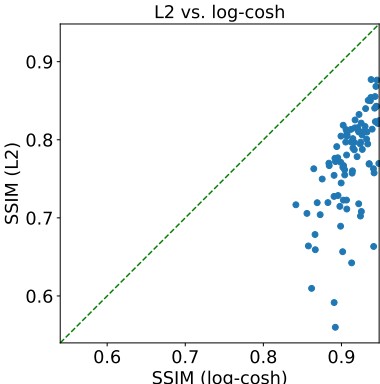 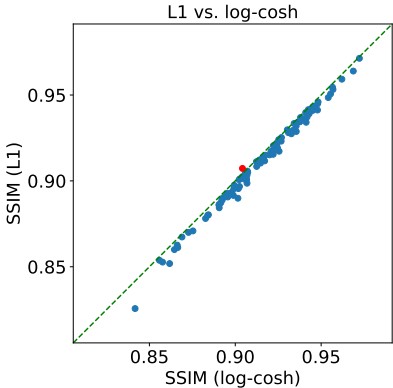

Figure 10: SSIM index of images restored with different loss. Each point corresponds to a restored image. Larger SSIM index is better. The green line indicates equal SSIM index. The log-cosh loss fails to outperform the $L_1$ loss in only 1 out of the 100 tested images as highlighted by red point

# D   APPLYING LOG-COSH TO IMAGE RESTORATION TASKS

It is promising to apply the log-cosh loss to various image restoration applications such as denoising, deblurring, demosaicking, and super-resolution. Here we provide an example of text removal, and show that the log-cosh loss consistently outperforms the $L_2$ and $L_1$ loss. We use a recently proposed denoising framework called noise2noise Lehtinen et al. (2018), which learns to restore good images by only going through bad ones. The network architecture is based on the idea of convolutional auto-encoder, consisting of convolutional and deconvolutional layers. The training dataset consists of 291 images widely used (e.g. in Schulter et al. (2015); Kim et al. (2016)). Among these, 200 images of the training set come from the Berkeley Segmentation Dataset Martin et al. (2001) and the rest 91 images come from Yang et al. (2010). Following Lehtinen et al. (2018), we add texts to the images and let the model learn to remove these texts. After training, we test the trained model on the Urban 100 dataset Huang et al. (2015) which consists of 100 high resolution images. We use the SSIM index Wang et al. (2004) which is often used as a measure to evaluate the similarity between the restored images and the target images in the denoising tasks. We compare the squared $L_2$ loss and the $L_1$ loss to the log-cosh loss, and the results are shown in Fig. 10. We can see that the log-cosh loss consistently outperforms both the squared $L_2$ loss and the $L_1$ loss. Actually, the log-cosh loss fails to outperform the $L_1$ loss in only 1 out of the 100 tested images, illustrated by the red point in Fig. 10.

We list some restored images in Fig. 11 to show how the log-cosh loss stands out. The $L_2$ loss may lead to a severe color shift, because the $L_2$ loss encourages the outputs to shift toward the mean color of the noisy images, which is locally distorted by texts. The $L_1$ loss encourages to take the median and does not suffer from color shift, but severe artifacts can appear, as highlighted by red boxes in Fig. 11. The log-cosh loss releases the problems by averaging observations around the median. It behaves like the $L_1$ at the beginning of training when the error is large, thus avoiding the color shift. In addition, when the error is small, The log-cosh loss behaves like the $L_2$ and encourages to smooth the color by taking the mean, thus reducing the artifacts introduced by the $L_1$ loss. As highlighted by red boxes, in the results of the log-cosh loss, the artifacts are removed or sometimes still exist but their color is much closer to the background.

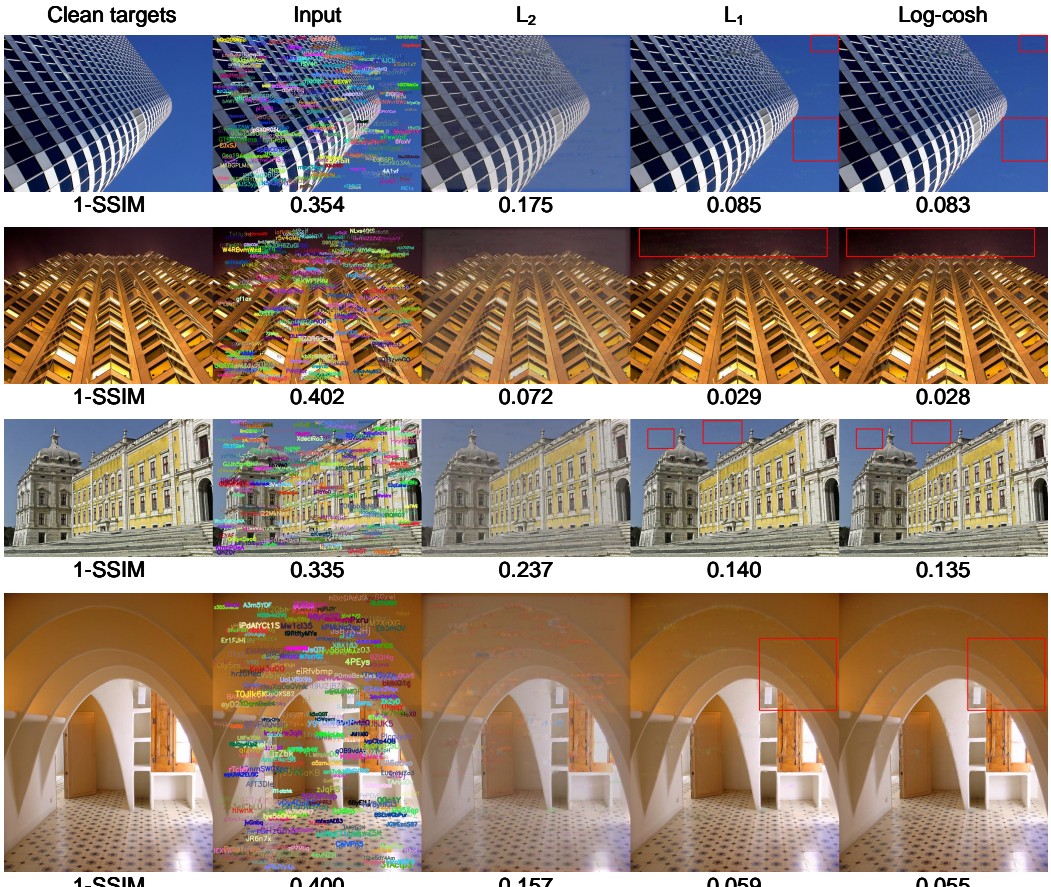

| Clean targets | Input | $L_2$ | $L_1$ | Log-cosh |
|---|---|---|---|---|
| 1-SSIM | 0.354 | 0.175 | 0.085 | 0.083 |
| 1-SSIM | 0.402 | 0.072 | 0.029 | 0.028 |
| 1-SSIM | 0.335 | 0.237 | 0.140 | 0.135 |
| 1-SSIM | 0.400 | 0.157 | 0.059 | 0.055 |

Figure 11: Text removal results of models trained with the squared $L_2$, $L_1$ and log-cosh loss. We list 1-SSIM here to show the difference between the restored images and the target images. The results of the $L_2$ loss suffers from severe color shift. The results of the $L_1$ loss suffer from severe artifacts. Both problems are released by the log-cosh loss.

