# OpenReview forum: "Log Hyperbolic Cosine Loss Improves Variational Auto-Encoder"
_ICLR.cc/2019/Conference_

### Official Review · AnonReviewer1 · 2018-11-01
**Interesting trick which can be used across various auto-encoder models with a rather convincing experimental evidence.**

**Rating:** 5
**Confidence:** 4

**Review:**

The paper proposes a new loss function which can be used in the reconstruction term of various auto-encoder architectures. The pixel-wise cost function \ell(X, X') = f(X - X'; a) is defined for pairs of two input images X and X' and has one positive real-valued hyperparameter a. For small values of t the function f(t; a) behaves like a quadratic function, while for large t it behaves like |t|. As a consequence, it is smooth, everywhere differentiable (like L2) while not penalizing outliers too hard (like L1). The authors present several experiments conducted on MNIST and Celeba datasets, demonstrating that a simple change of a conventional pixel-wise squared L2 distance with the proposed log-cosh cost function improves the FID scores of generated samples as well as the visual quality of reconstructions (including "the sharpness").

I would say this is clearly an empirical study (even though the authors claim they provide "theoretical justifications", they are rather hand wavy), which is not a bad thing in this case. The message of the paper is very clear and I think the authors did a good job in selling their point. The main (and, perhaps, the only) contribution is the proposal to use the log-cosh function as the reconstruction cost. And this proposal is well justified by the set of experiments.

However, there are several major issues:
(1.1) The objective functions reported in appendix A.1 corresponding to WAE have in fact nothing to do with WAE. In WAE the regularizer penalizes the divergence between the prior distribution p(a) and *the aggregated posterior* distribution \int_x q(z|x) p(x) dx. In other words, D_MMD(q(z|x) || p(z)) in Eq. 8 should be replaced with D_MMD(\int_x q(z|x) p(x) dx || p(z)) in order to result in the WAE model. In summary, if the authors indeed used objectives reported in Eq. 8 of Appendix A, they were actually not using WAE but rather some other sort of regularized auto-encoders, which in a way are quite similar to VAEs.
(1.2)  I am surprised to see the reported FID scores for the Celeba data set. Having worked with this data set myself in combination with VAEs and WAEs, I am impressed with the extremely low FID scores: 46 for the vanilla L2 VAE and 30 for the L2 WAE. Note that while in the appendix the authors say they follow the architectural choices provided in [1] while performing the "L2 WAE Celeba" experiment, the authors arrive at FID=30 compared to FID=55 reported in the "Wasserstein Autoencoders" paper. Also, based on my experience, achieving FID=46 on CelebA with a vanilla VAE is very impressive. Note that the authors use 10^4 of samples to evaluate the FID scores, which is exactly the same as in [1]. This size is known to be large enough to reduce the variance of FID, so the difference (55 - 30) can not be explained by the fluctuations of FID. Therefore, I ask the authors to (anonymously) share the code and/or checkpoints of the 2 particular trained models: L2 VAE and L2 WAE trained on Celeba.

Other comments:
(2.1) Note that the reconstruction cost function in VAE should be normalized for every value of the code Z, as it corresponds to the logarithm of the likelihood (density) function -log p(X|Z). L2 and L1 costs both correspond to the well known likelihood (decoder) models (Gaussian and Laplace). However, it is hard to say what decoder model (what type of conditional distribution p(X|Z) ) would give rise to the proposed log-cosh function. In particular, the normalizing constant is not known and may depend on Z. In other words, by exchanging the L2 cost with the log-cosh loss in the VAE one looses the theoretical guarantees supporting VAE, including the fact that the objective is the lower bound on the marginal log likelihood. While this is not necessarily a problem (unless one uses the value of the objective as the bound on the marginal log likelihood, which is not the case in this paper), I would suggest mentioning it. Notice that, for instance, in WAE this problem does not appear, as the reconstruction term there does not involve any likelihood functions and thus does not need to be normalized.
(2.2) In Figure 2 I don't see why the authors did not highlight bad samples in the second row corresponding to their proposed method? I see many badly looking images there. Say, (4, 9) in VAE (MLP) and (8, 9) in VAE (Conv) and (6, 1) in WAE (MLP) and (2, 10) in WAE (Conv), where (i, j) means i-th row, j-th column, indexing starting from 1.
(2.3) How would the Huber loss perform and how does it compare to the proposed loss?

[1] Wasserstein Autoencoders. Tolstikhin et al., ICLR, 2018.

---

> ### Comment · AnonReviewer1 · 2018-11-23
> **Authors do not report standard evaluation metrics**
>
> Based on the latest authors' reply I need to conclude that I am not changing my score.
> Instead of reporting the FID scores following a standard implementation widely accepted these days,
> it turns out the authors were using their own classifier trained on CelebA to embed the pictures. It is
> my fault that initially I did not notice this fact, even though it is indeed explicitly stated in the paper.Even
> though the same evaluation process was applied to all the methods (and in this way the authors were
>  "fair"), the properties of this reported metric are completely unknown and thus the numbers can not
> be viewed as reliable for any sort of conclusions.

---

### Official Review · AnonReviewer3 · 2018-11-02
**well written, but contribution is unclear and evaluation insufficient**

**Rating:** 4
**Confidence:** 4

**Review:**

+ well written and explained
+ well motivated
- Unclear if it helps prevent blurry images
- No comparison to similar loss functions or different tasks

The paper is well written and very easy to follow. I really liked the introduction as it reads well and clearly motivates the problem. The authors correctly highlight the two major issues in VAE's and propose to solve one of them (the reconstruction loss).

One of the major issues is that the proposed solution does not solve the problem of blurry images. There are two reasons why a generative model might produce a blurry output with an L2 (or L1) loss:
 1. The training data is noisy and the best fitting generation will average this noise. This is the issue the authors propose to solve.
 2. A much larger issue is that the generative model might be uncertain about the spatial location of objects. Here, again a blurry generation is the most optimal output. However unlike (1.) a different loss, like L1 or log-cosh, does not address this issue. The blurriness primarily comes from the element-wise nature of the loss function. Hence simply making the loss robust to outliers (in terms of color values) is not enough.

The second major issue in the paper is a lack of comparison to other alternative loss functions. As the authors mention in their intro, there has been a host of proposed solutions to the blurry generation: optimizing L1, SSIM, a perceptual loss (e.g. VGG features) and many more. However, the authors do not compare to any of them, and simply setup their main comparison with a squared L2 loss. I would expect the authors to at least compare to other simple loss functions. At a minimum a comparison should contain:
 * L2 (not squared)
 * L1
 * SSIM

In my view the weaknesses currently outweigh the strength of the submission.

---

### Official Review · AnonReviewer2 · 2018-11-03
**Missing theoretical analysis of Log-Cosh Loss**

**Rating:** 4
**Confidence:** 4

**Review:**

This paper proposes to change the L2 norm of loss function of VAE into hyperbolic cosh function. The idea  and presentation are clear and straightforward. However, the used cosh function does not convince me since when t=a, f(t,a) will still be very large! Also, they will grow fast with exp|at|. The authors are encouraged to provide more detailed proofs for the advantages of cosh function.

Apart from the cosh loss, the Huber loss is well-known robust loss function used in statistics and many computer vision applications, and it has the similar properties of cosh function. I feel surprised that the authors do not aware this and do not compare it in experiments.

The introduction is a bit confusing. GAN is an implicit generative model as it does not have any explicit density form, but the likelihood and prior of vanilla VAE are Gaussian. I am not clear what is the motivation to introduce the cosh loss function.

If the authors aim to improve the generative quality, there are several works, such as using PixelCNN or other advanced likelihoods, improve the VAE. Besides these, recently MAE uses mutual information as the regularization to improve the quality.

Overall, this work does not convey any theoretical analysis and significant results over state-of-the-art.

---

### Author Response · Authors · 2018-11-19
**Thanks for your comments**

Thank you very much for all your insightful comments. We have compared the proposed loss with squared L2 (and L1 in the appendix) on different architectures, demonstrating significantly improvement of the performance of VAE. Still, comparisons with other common losses such as Huber loss, SSIM and perceptual loss are missing. We do need stronger theoretical and experimental results to justify the advantages and practical usages of the proposed log-cosh loss. Thank you again and we should polish this work further.

Moreover, I would like to clarify two additional issues argued by reviewer1.
(1.1) In the implementation, we do use D_MMD(\int_x q(z|x) p(x) dx || p(z)), which is the same as WAE [1].
(1.2) For the calculations of FID scores. We trained a new CNN classification model on CelebA, which achieved a test accuracy of 91.5% on average over human face related features, including gender, smile, make up, etc. The FID was then evaluated on a layer of our own CNN model rather than the default Inception network. Thus, the FID sores might seem different to [1]. We think that a model trained specifically for the CelebA dataset actually distills features most related with face than models trained on other datasets, such as ImageNet. The calculations of FID were actually fair since we compared all methods based on the same CNN. As required by the reviewer, we share the code for L2 VAE and L2 WAE (and also the CNN classification model) trained on CelebA via an anonymous google drive link: https://drive.google.com/drive/folders/1SGE4ghCok-MC6kNPUFW1VH1m8QU6UOum?usp=sharing

[1] Wasserstein Autoencoders. Tolstikhin et al., ICLR, 2018.

---

### Meta-Review · Area_Chair1 · 2018-12-14
**Well-written and promising method, but some remaining issues around theoretical justification and experimental metrics.**

**Confidence:** 5
**Recommendation:** Reject

**Metareview:**

The reviewers agree that the paper is well-written, and they all seem to like the general idea. One of the earlier criticisms was that you did not compare against other robust loss functions, but you have partially rectified that by comparing to L1 in the appendix. As per the request of reviewer 2 I would also compare to the Huber loss.

One remaining concern is the lack of theoretical justification, which could help address the comment of reviewer 3 regarding blurry images from location uncertainty. The other concern is that you should compare your method using FID scores from a standard implementation so that your numbers are comparable to other papers. Some of the reviewers were impressed, but confused by your relatively low scores.